# Eosinophilic Myocarditis: From Bench to Bedside

**DOI:** 10.3390/biomedicines12030656

**Published:** 2024-03-14

**Authors:** Francesco Piccirillo, Sara Mastroberardino, Vincenzo Nafisio, Matteo Fiorentino, Andrea Segreti, Annunziata Nusca, Gian Paolo Ussia, Francesco Grigioni

**Affiliations:** 1Fondazione Policlinico Universitario Campus Bio-Medico, Via Alvaro del Portillo 200, 00128 Roma, Italy; sara.mastroberardino@unicampus.it (S.M.); vincenzo.nafisio@unicampus.it (V.N.); matteo.fiorentino@unicampus.it (M.F.); a.segreti@policlinicocampus.it (A.S.); a.nusca@policlinicocampus.it (A.N.); g.ussia@policlinicocampus.it (G.P.U.); f.grigioni@policlinicocampus.it (F.G.); 2Research Unit of Cardiovascular Sciences, Department of Medicine and Surgery, Università Campus Bio-Medico di Roma, Via Alvaro del Portillo 21, 00128 Roma, Italy; 3Department of Movement, Human and Health Sciences, University of Rome “Foro Italico”, Piazza Lauro de Bosis 15, 00135 Roma, Italy

**Keywords:** myocarditis, eosinophilic myocarditis, eosinophilic granulomatosis with polyangiitis, hyper-eosinophilic syndrome, endomyocardial fibrosis, cardiomyopathy

## Abstract

Myocarditis is a polymorphic and potentially life-threatening disease characterized by a large variability in clinical presentation and prognosis. Within the broad spectrum of etiology, eosinophilic myocarditis represents a rare condition characterized by eosinophilic infiltration of the myocardium, usually associated with peripheral eosinophilia. Albeit uncommon, eosinophilic myocarditis could be potentially life-threatening, ranging from mild asymptomatic disease to multifocal widespread infiltrates associated with myocardial necrosis, thrombotic complications, and endomyocardial fibrosis. Moreover, it could progress to dilated cardiomyopathy, resulting in a poor prognosis. The leading causes of eosinophilic myocarditis are hypersensitivity reactions, eosinophilic granulomatosis with polyangiitis, cancer, hyper-eosinophilic syndrome variants, and infections. A thorough evaluation and accurate diagnosis are crucial to identifying the underlying cause and defining the appropriate therapeutic strategy. On these bases, this comprehensive review aims to summarize the current knowledge on eosinophilic myocarditis, providing a schematic and practical approach to diagnosing, evaluating, and treating eosinophilic myocarditis.

## 1. Introduction

Myocarditis is an inflammatory heart disease characterized by considerable variability in clinical presentation and evolution [1]. The presentation of myocarditis is strictly connected to the underlying pathogenic factors and inflammatory conditions, which could damage the myocardial tissue with varying severity [2]. Eosinophilic myocarditis (EM) is a rare nosological entity characterized by a mild-to-severe eosinophilic infiltration in the myocardium, usually associated with peripheral eosinophilia. EM may be related to hypersensitivity reactions, immune-mediated disorders such as eosinophilic granulomatosis with polyangiitis (EGPA), hyper-eosinophilic syndrome (HES), myeloproliferative diseases, infections, and cancer. Moreover, in a not negligible number of cases, the underlying cause remains unknown. The clinical presentation largely varies, ranging from pauci- or asymptomatic disease to acute fulminant myocarditis or chronic restrictive cardiomyopathy [3]. A specific therapy for eosinophilic myocarditis is not supported by large-scale clinical trials; current treatments include both pharmacological and non-pharmacological therapies for heart failure and immunosuppressive therapies for advanced stages and/or immune-mediated disorders [4]. Albeit difficult to determine, the natural history of EM may range from complete recovery to evolution in cardiomyopathy or death due to severe systolic dysfunction or ventricular arrhythmias in their fulminant form.

On these bases, this comprehensive review aims to summarize the current literature on eosinophilic myocarditis, providing schematic and practical information on diagnosis, evaluation, and treatment and improving awareness about this nosological condition.

Details regarding research methods are described in Appendix A.

## 2. Epidemiology

Myocarditis is a significant challenge for clinicians due to its diagnostic complexity and various clinical presentations. An accurate definition of the incidence of eosinophilic myocarditis is complex, primarily because of its potentially subtle symptomatic presentation, often leading to post-mortem biopsy diagnoses [5]. Indeed, several studies estimated the presence of eosinophilic myocarditis (EM) in 20% of hearts explanted for transplantation and 0.5% of non-selected autopsies [6,7,8]. Moreover, the endo-myocardial biopsy, which represents the diagnostic gold standard, is not always feasible due to its invasiveness, and it could sometimes be non-diagnostic because of the multifocal infiltrative nature of myocardial engagement [5,6,9]. While myocarditis is a male-predominant disease with incidence decreasing with age [10], EM was more prevalent in Caucasians with a mean age of 41 years (in patients with a histological diagnosis) and 46 years (without a histological diagnosis), with no statistically significant differences between sexes [3]. Despite the limited number of case reports in the literature, a systematic review conducted by Brambatti et al. on 264 patients (176 with a histological diagnosis of EM) revealed that eosinophilic myocarditis is associated with an associated systemic disorder in 64% of cases [3]. Specifically, the most reported cause of EM was hypersensitivity (34% of patients), followed by eosinophilic granulomatosis with polyangiitis (about 13%) and hyper-eosinophilic syndrome (approximately 8%) [3]. Other causes were infections (5%), mainly associated with *Toxocara canis*, pregnancy-related EM, malignancies, and other immune disorders [3]. Furthermore, hyper-eosinophilic syndrome (HES) myocarditis accounts for 8.4% of eosinophilic myocarditis cases, with an estimated incidence of 0.036 per 100,000 [11]. EM resulted in idiopathic or undefined in 35% of cases [3]. No epidemiological information can be accurately defined, given the limited cases reported in the literature regarding less frequent forms of EM, such as those related to infections, pregnancy, malignancy, and toxic conditions.

## 3. Etiology

The exact etiopathogenetic mechanism of eosinophilic myocarditis (EM) is not yet fully understood, but genetic, environmental, and immunological aspects are likely involved. Much remains to be learned about the genetic elements, although some progress is being made. Firstly, Barin et al. attempted to analyze the development of EM, considering the lack of interferon-γ (IFN-γ) and interleukin-17A (IL-17A) in murine models [12]. IFNγ−/−IL17A−/− mice developed a rapidly fatal EM, demonstrating the importance of these mediators in the host inflammatory response. In another study conducted by Yoshida T et al., Bcl-6-deficient mice showed worse progression of eosinophilic inflammation, proving the protective effect of this inflammatory mediator [13]. Similarly, some studies have been carried out in humans to identify differences in the HLA haplotype in the inflammatory response, which could explain a predisposition to the development of eosinophilic granulomatosis with polyangiitis (EGPA), but the results were not very promising. It appears that the HLA-DRB1*07 and HLA-DRB4 haplotypes are genetic risk factors for EGPA, but more research is needed [14,15].

The leading causes of EM can be divided into three categories: toxic, immunologic, and infectious [16]. Drugs more frequently associated with EM hypersensitivity are antibiotics (36.5%, mainly minocycline and β-lactam), central nervous system agents (21.1%, primarily clozapine and carbamazepine), anti-inflammatory (largely indomethacin), diuretics, vaccines (7.7%, mostly tetanus toxoid and smallpox), and antitubercular agents (1.9%) [17,18,19]. Furthermore, rare cases of EM correlated with prolonged exposure to high doses of dobutamine have been described in the literature, even in the absence of peripheral eosinophilia [20,21]. Indeed, in cardiac transplant candidates, EM could represent an occasional finding on post-transplant histological examination, particularly in patients with long-term dobutamine treatment [22]. As mentioned above, EM could be related to vaccines. Specifically, rare cases of EM after COVID-19 vaccination have been recently described, but only one patient needed mechanical circulation support [23,24,25]. In addition, EM has been observed in children after a single vaccination with meningococcal C or hepatitis B [26].

Eosinophilic granulomatosis with polyangiitis (EGPA, also known as Churg–Strauss Syndrome) is a rare type of vasculitis with eosinophilic granulomatous inflammation and necrosis of small-medium blood vessels, associated with c-ANCA positivity in 40–60% of cases [27,28]. Clinically, EGPA is characterized by pulmonary infiltrates (often migrating), poly or mononeuropathies, polyserositis, and cardiac and renal involvement. The diagnosis is based on the following criteria coded by the American College of Rheumatology (a score ≥ 6 is required): obstructive airway disease (+3), nasal polyps (+3), mononeuritis multiplex (+1), blood eosinophil count ≥ 1 × 10^9^/L (+5), extravascular eosinophilic-predominant inflammation on biopsy (+2), positive test for cytoplasmic antineutrophil cytoplasmic antibodies (cANCA) or antiproteinase 3 (anti-PR3) antibodies (−3), and hematuria (−1) [27]. Cardiac involvement develops in about 60% of patients, with a poor prognosis in most cases (50% mortality), and it is more common in ANCA-negative patients. Specifically, patients with EGPA may show eosinophilic myocarditis, heart failure, pericarditis, arrhythmias, coronary arteritis, valvulopathy, or intracavitary cardiac thrombosis [29]. Interestingly, inflammatory cardiac involvement could also be observed in asymptomatic patients and was associated with more extensive disease activity, a higher eosinophil count, and ANCA negativity [29]. Indeed, lower overall rates of cardiac involvement in EGPA patients could be related to cardiac assessments performed only on symptomatic patients, underestimating the real prevalence of heart disease [30]. Thus, a systematic assessment to detect cardiac disease could be performed on all EGPA patients, regardless of the presence of symptoms [29].

Hyper-eosinophilic syndrome (HES) is characterized by an absolute eosinophil count above 1.5 × 10^9^/L for more than six months, damaging the bone marrow, nervous system, and heart. HES could be idiopathic, when the hyper-eosinophilia has no cause, or it could be secondary, which is typically associated with myelo- or lympho-proliferative hematological disease [18]. Mainly, HES includes Davies’ endomyocardial fibrosis and Loffler’s myocarditis. Davies’ endomyocardial fibrosis, also known as tropical endomyocardial fibrosis (TEMF), is a restrictive cardiomyopathy of unclear etiology that is endemic in Africa, South America, and India and extremely rare in other latitudes [31]. The exact etiological and pathological mechanisms are unclear. Firstly, a relationship among fibrotic damage, eosinophil toxicity, and serotonin and its metabolites was hypothesized, given the similar cardiac damage observed in TEMF, carcinoid syndrome, and Loffler’s myocarditis [32]. Moreover, the specific geographical distribution of the disease suggested a viral or helminthic (probably filariasis) origin [32]. Unfortunately, the few studies that have been carried out so far have not yet demonstrated the existence of an intermediate stage of eosinophilic myocarditis, which may precede the development of fibrosis, thus allowing a timely diagnosis and treatment. Moreover, Loffler’s syndrome is a form of eosinophilic lung disease characterized by absent or mild respiratory symptoms involving a dry cough, transient and migratory pulmonary opacities, and peripheral blood eosinophilia. It could result from parasitic infections, which may be the cause, albeit sometimes no identifiable pathogen is found [33]. The disease involves the heart, causing endocarditis and eosinophilic myocarditis, which, over time, leads to endomyocardial fibrosis and, finally, restrictive cardiomyopathy [5].

Moreover, EM could develop due to parasitic infestations due to the related persistent hyper-eosinophilia. The most commonly involved helminths infections are due to *Toxocara canis* (the most frequent one), *Trichinella spiralis*, *Entamoeba fragilis*, *Isospora belli*, and protozoa, like *Trypanosoma cruzi* and *Toxoplasma gondii* [5,9,17]. Specifically, *Toxocara* spp. may infect dogs and cats, which shed the eggs in their feces; humans can ingest contaminated vegetables, and thus, larvae penetrate the intestine, reaching the liver via the portal vein and then the systemic circulation [34]. The TES-IgG ELISA is recommended when EM in toxocariasis is suspected [17]. Furthermore, EM could be associated with a viral infection. In particular, a single case of a 17-year-old young man with autopsy findings of EM and COVID-19 positivity in the absence of other possible causes of eosinophilic myocarditis has been described so far [35]. Rarely, the human immunodeficiency virus (HIV) may cause EM due to the persistent elevation of eosinophil counts, which could infiltrate and damage cardiac muscle [36]. Interestingly, high eosinophil levels correlate with low CD4 levels in HIV-positive patients [37,38]. EM may be observed in patients with neoplastic disorders, such as T-cell lymphomas and some carcinomas involving lung and biliary cancers [39]. In addition, rare cases of EM have been associated with pregnancy and the peripartum, but the exact etiopathological mechanism remains to be fully elucidated [40,41].

The different etiologies of EM with related clinical conditions are schematized in Table 1.

## 4. Pathophysiology

Although it has a different underlined etiology, eosinophilic myocarditis (EM) is characterized by varying degrees of eosinophil infiltration, which are often associated with peripheral eosinophilia [42,43,44]. From a histological point of view, the progression of the disease could be categorized into stages, which may overlap (Figure 1). Firstly, the initial phase of acute necrosis corresponds to the extensive infiltration of eosinophils into the cardiac tissue [45]. The cellular damage is promoted by the degranulation of eosinophils, which leads to the release of major essential proteins and an elevated expression of granulocyte/macrophage colony-stimulating factor, interleukin-3 (IL-3), and IL-5 receptors [46,47]. Specifically, IL-5, along with eosinophilic cationic protein (ECP), contributes to increased recruitment and degranulation of eosinophils [16]. Cell damage occurs via two main mechanisms: the first involves the eosinophilic cationic protein, which promotes the release of histamine and tryptase through cardiac mast cells; secondly, the primary essential protein increases cell permeability and inhibits the mitochondrial respiratory mechanism [16]. In addition to direct damage, eosinophils contribute to microvascular damage associated with the hyperactivation of endogenous coagulation systems and autoimmune activation [18,48]. Moreover, several studies have shown that eosinophils release different molecules involved in blood coagulation as tissue factors, promoting a hypercoagulable state [49,50]. The raised coagulation activity leads to a subsequent thrombocyte phase, predominantly apical, which carries a significant risk of embolization [18]. The final phase of the disease is the fibrotic stage, which can affect both valve and heart wall structures, often requiring surgical intervention [45]. Interestingly, the mechanisms described are not entirely independent, as necrosis, thrombosis, and fibrous scarring could coexist, leading to an early interstitial fibrotic phase. In addition, scar tissue and fibrosis could occur after cell necrosis and apoptosis [51]. Thus, it could be hypothesized that eosinophilic myocarditis, Loeffler’s endocarditis, Davis disease, and myocardial fibrosis represent different stages of a single disease induced by eosinophil-mediated cardiac injury. In addition, a study conducted in mice showed the importance of eotaxins and CCR3 for eosinophil migration and localization to the heart [52]. Thus, albeit no treatments have been studied in humans for eosinophilic myocarditis, targeting eotaxins or CCR3 could be a turning point in preventing eosinophil-mediated heart damage. Similarly, mice deficient in interferon-γ and IL17A developed a rapidly fatal EM, thus suggesting protective effects of INF-γ and IL17A in eosinophilic heart disease [12].

Moreover, specific features can be identified in the associated ANCA vasculitis and Loeffler’s endocarditis. Both diseases are associated with inflammation and necrosis, affecting the entire wall thickness. Additionally, vascular involvement with the formation of granulomas in small vessels is possible in ANCA-associated diseases [16]. Specifically, the vascular districts most affected are the small-caliber veins and small- to medium-caliber arteries, with a rare involvement of medium- to large-caliber arteries [16]. Eosinophilic hypersensitivity myocarditis, in contrast, exhibits the absence of granulation areas and shows a mixed infiltrate, including eosinophils, lymphocytes, and histocytes, with peri-vascular and interstitial localization [16].

## 5. Clinical Manifestations

Clinical manifestations of eosinophilic myocarditis (EM) are generally nonspecific and could vary from mild asymptomatic disease to life-threatening cardiogenic shock and ventricular arrhythmias. Usually, EM onset may include acute chest pain or tightness, shortness of breath, and elevated creatine kinase MB and troponin levels [16]. Interestingly, dyspnea and chest pain represent the most frequent symptoms at presentation, while about 20% of patients present with misleading manifestations, including asthenia, nausea, and myalgia [3]. Fulminant myocarditis represents an acute condition characterized by hemodynamic instability and malignant arrhythmias due to severe inflammatory damage [1]. Chronic forms are often the result of a previously unrecognized acute phase and usually occur as progressive heart failure, characterized by a recent onset of systolic dysfunction of the left ventricle or unexplained ventricular arrhythmias, even in the absence of systolic dysfunction [3]. In addition, EM could show symptoms related to different organs involved, including fever, cough, and pharyngodynia (Figure 2) [17].

In EGPA-associated EM, the clinical presentation is characterized by multi-organ involvement that represents the pathophysiological expression of the blood eosinophilia, eosinophilic infiltrates, and necrotizing vasculitis of small vessels [53]. Beyond cardiac involvement, which includes myocarditis, pericarditis, coronary vasculitis, and heart failure, and could also be asymptomatic, non-cardiac manifestations are mainly represented by respiratory symptoms, such as bronchial asthma, usually associated with chronic rhinosinusitis, nasal polyposis, and pulmonary manifestations due to the presence of migratory infiltrates in the upper airways and lungs [54]. In addition, peripheral neuropathy is observed in 75–80% of cases, generally rapid and worsening, with peroneal and internal popliteal nerves most commonly involved [54]. Dermatological manifestations include palpable purpura with scalp nodules, urticarial rashes, skin infarcts, and livedo reticularis [54]. The kidney and gastrointestinal system could also be affected, including abdominal pain, nausea, diarrhea, and ANCA-related nephritis [54].

Generally, cardiac involvement in hyper-eosinophilic syndrome (HES) is not always symptomatic, although it could manifest both as an acute or a chronic injury and suggests a poor prognosis [55]. The most common manifestations of HES are dermatologic, with skin lesions/rash and pruritus [56]. Pulmonary and gastrointestinal symptoms are observed in about 25% of patients [56]. Cardiac manifestation in a chronic setting includes signs and symptoms of heart failure, such as exertional or worsening dyspnea, peripheral edema, chest discomfort, and asthenia [17].

## 6. Diagnosis

The early diagnosis of eosinophilic myocarditis (EM) is essential to improving the prognosis, albeit the underlying variety of rare and complex clinical conditions could delay the diagnosis. The first step is to consider haemato-chemical investigations. An increase in myocardial markers (especially troponin I and NT-proBNP), inflammatory indices (erythrocyte sedimentation rate and C-reactive protein), and leukocytes, especially in their eosinophilic component, could be expected [17]. Interestingly, EM is not always associated with peripheral hyper-eosinophilia. However, patients without eosinophilia at admission could develop peripheral eosinophilia during hospitalization [3,57]. Thus, white blood cell exams, including eosinophil count, should be repeated during hospitalization to avoid misdiagnosis. As previously mentioned, cases of EM associated with dobutamine in the absence of peripheral hyper-eosinophilia have been described in the literature [20]. In other cases, during the acute stage of EM, there is a paradoxical phase in which there is no peripheral hyper-eosinophilia because of the migration of eosinophils into the tissues and the simultaneous eosinopoiesis in the bone marrow [58]. In cases of EGPA suspicion, anti-neutrophil cytoplasmic antibodies (ANCA) should be sought, even if these are generally absent in EGPA with cardiac involvement [59].

The second step is the standard 12-lead electrocardiogram (ECG), albeit myocarditis has no pathognomonic ECG patterns [1]. ECG could show sinus tachycardia, atrioventricular or bundle branch block, ST-wave, and T-wave changes [1]. Cardiac inflammation and scarring could induce different types of arrhythmias involving sinus tachycardia and complex ventricular arrhythmias [60]. Although only a few patients develop malignant arrhythmias during the acute phase, hypersensitivity EM is most associated with the risk of developing ventricular tachycardia and/or fibrillation [3]. Moreover, bradyarrhythmia could also occur [15]. Diffuse ST segment changes, both supra- and sub-segment, may be observed, especially in the case of related pericarditis, miming an acute coronary syndrome [3,15]. In addition, abnormalities in repolarization include negative, isoelectric, triphasic, or biphasic T waves. Finally, in the case of concomitant pericarditis with consensual pericardial effusion, a reduction in QRS voltages may be observed [15].

Coronary angiography or coronary CT are recommended in particular in acute coronary syndrome-like manifestations, including ST segment elevation, raised cardiac troponins, and wall motion abnormalities [4,16]. Although not mandatory, a chest x-ray and eventually a chest CT scan may be performed as part of the diagnostic workup. They may show cardiomegaly, pulmonary infiltrates, pleural effusion, and pulmonary and cardiac thrombosis [18]. Indeed, eosinophils interact with platelets via the major basic protein contained in their granules, which is a potent platelet stimulator; therefore, the migration of eosinophils into tissues promotes inflammation and the development of thrombosis [61,62]. In addition, activated eosinophils express tissue factors that are thrombogenic per se [18].

An echocardiogram is mandatory, although the non-invasive gold standard for diagnosis is cardiac MRI. The echocardiogram allows an initial differential diagnosis with other possible causes of acute heart failure (including cardiomyopathies, genetic cardiopathies, and valvulopathies), the determination of cardiac function, the assessment of concomitant valvular defects, the possible pericardial involvement, and the presence of an associated pericardial effusion [1,3]. The presence of an intracavitary thrombus could be assessed, given the pro-thrombotic status, as previously said [3,17]. Interestingly, in the last few years, several studies have described acute valvulopathy secondary to EM, including a case of severe mitral stenosis secondary to EGPA that resolved after pharmacological treatment [63]. In addition, mitral and tricuspid regurgitation are more common echocardiographic findings, probably related to inflammatory damage [64]. Finally, a trans-thoracic echocardiogram is crucial to assess the response to therapy and the progression of the disease, which could result in dilated or restrictive cardiomyopathy, as observed in Loeffler’s myocarditis [17].

Cardiovascular magnetic resonance (CMR) provides tissue evaluation of the myocardium and represents the non-invasive diagnostic gold standard. In 2009, the Lake Louise Criteria for diagnosing myocarditis on CMR were formulated, with a sensitivity of 80% and a specificity of 87% [65]. Mainly, T1- and T2-weighted images, in addition to early and late gadolinium enhancement acquisitions, define specific markers of tissue damage: hyperemia and capillary leakage, necrosis and fibrosis, and intracellular and interstitial edema [65]. Specifically, in the acute phase, CMR assesses the presence of edema, which appears hyperintense on T2-weighted sequences. Nevertheless, turbo inversion recovery magnitude imaging (TIRM) is more sensitive [16]. Moreover, in a chronic setting, the presence of fibrosis or necrosis, which typically has a patchy and subendocardial distribution, is detectable on late gadolinium enhancement (LGE) sequences. This specific distribution pattern could provide a differential diagnosis of post-ischemic fibrosis/necrosis, which is typically trans-mural and distributed along the territory of one or more coronary arteries [16]. Although no specific patterns are described, CMR in patients with EM typically shows a subendocardial LGE distribution, particularly in EM related to EGPA (Figure 3) [3].

Myocardial involvement in ANCA-associated vasculitis, such as EGPA, could be silent, presenting with no symptoms, normal ECG, and preserved left ventricular ejection fraction on echocardiogram until the development of dilated cardiomyopathy and end-stage heart failure [66,67]. CMR could detect pathophysiologic phenomena, including acute/chronic inflammation, coronary macro- and micro-circulation abnormalities, and/or small vessel vasculitis, occurring during systemic vasculitis and also in subclinical cases [68,69]. Interestingly, a study conducted by Greulich et al. found that ANCA-associated vasculitis patients with high disease activity (Birmingham Vasculitis Activity Score > 5) showed diffuse fibrotic and inflammatory myocardial changes, including higher values for native T1, extracellular volume (ECV), and T2, compared to controls, regardless of LGE [70]. Most patients were non-symptomatic, with normal ECG and echocardiography, suggesting CMR with mapping techniques could detect early and subtle diffuse myocardial fibrosis in patients with otherwise normal cardiac evaluation [70]. Similarly, albeit systemic involvement in EGPA could lead to an underestimation of signs related to EM, CMR may demonstrate focal replacement and diffuse interstitial myocardial fibrosis also in patients with stable EGPA [71]. In addition, CMR plays a fundamental role in risk stratification, as the cardiac involvement in EM related to EGPA is highly heterogeneous, so it should be considered as a spectrum of patterns that correspond to different clinical pictures. A retrospective study conducted by X. Liu et al. analyzed EGPA patients from 2012–2023, identifying three groups based on cardiac enzymes, cardiac magnetic resonance imaging, and endomyocardial biopsy results: eosinophilic myocarditis (EGPA-EM), chronic inflammatory myocarditis/cardiomyopathy (EGPA-ICM), and EGPA-control, which differ in terms of urgency, EGPA-related manifestations, eosinophil count, severity of cardiac injury, cardiac inflammation, and cardiac manifestations [72]. Specifically, EGPA-EM patients showed a significantly worse prognosis, with a death rate of about 15% and a 2-year event-free survival rate below 50%, underlying the importance of more aggressive treatment [72]. Finally, CMR could be useful to monitor therapeutic efficacy [73]. Indeed, a study conducted by Fijolek et al. on EGPA patients found that all individuals had cardiac involvement, with myocardial edema and perfusion defects detected in about 90% and 55%, respectively [73]. Control CMR showed improvement in about 80% of patients, a complete remission in 10%, and an evolution in global fibrosis in 35% of individuals [73]. These findings suggest the crucial role of CMR in EGPA, as it could detect patients with cardiac injury, define who needs combined therapy, and evaluate the therapeutic effect.

The invasive diagnostic gold standard for suspected myocarditis is endomyocardial biopsy (EMB). As assessed by Dallas histopathological criteria, myocarditis is defined as evidence of inflammatory infiltrates within the myocardium associated with myocyte degeneration and necrosis of non-ischemic agents [74]. Specifically, there are three scenarios in which EMB should be used: new-onset heart failure of <2 weeks’ duration with normal-sized ventricles or left ventricular dilatation associated with hemodynamic instability; new-onset heart failure of 2 weeks to 3 months duration associated with left ventricular dilatation and the development of new ventricular arrhythmias, or second- or third-degree atrioventricular block or failure to respond to decompensated heart failure therapy for 1–2 weeks; heart failure with dilated cardiomyopathy of any duration associated with suspected allergic reaction and/or eosinophilia [75]. However, unlike CMR, EMB is not very sensitive (about 50%), given the typically focal nature of inflammatory damage [16]. Moreover, acute myocardial infarction, left ventricular thrombosis, and ventricular aneurysm represent absolute contraindications to EMB.

Nevertheless, EMB represents the only method that is able to define the characteristics and histological subtypes of cardiac inflammation. Specifically, acute myocardial inflammation with interstitial eosinophilic infiltration could be detected through hematoxylin and eosin staining (Figure 4) [17]. In addition, EMB could show myocellular damage, which comprises a heterogeneous spectrum of histopathological changes, including necrosis, myo-cytolysis, cytoplasmic clarification/vacuolization, fragmentation, and loss of myocytes [7]. Interestingly, myo-cytolysis is associated with more severe interstitial and perivascular eosinophilia [7]. Specifically, myocyte necrosis is related to eosinophil degranulation and deposition of eosinophilic granule MBP, inhibited mitochondrial respiration, and raised cell membrane permeability [46]. Moreover, necrotizing eosinophilic myocarditis, a rare condition described only in a few case reports, is characterized and identified by a diffuse inflammatory infiltrate with predominant eosinophils associated with extensive necrosis [75]. Necrotizing eosinophilic myocarditis differs from HES-related EM as lesions result in diffuse rather than perivascular and interstitial, with prominent myo-cytolysis [75].

## 7. Therapy

Albeit a specific therapy for eosinophilic myocarditis (EM) is not supported by large-scale clinical trials, but the current treatments include both pharmacological and non-pharmacological therapy in uncomplicated and complicated EM manifesting with acute heart failure.

Table 2 summarizes therapeutic strategies in eosinophilic myocarditis.

Firstly, patients are advised to restrict physical activity during the acute phase of myocarditis and over the following six months [4]. Pharmacological therapy in uncomplicated forms includes chronic heart failure drugs such as low-dose beta-blockers, angiotensin-converting enzyme inhibitors/angiotensin receptor blockers, and aldosterone receptor antagonists, which could improve myocardial remodeling [4]. In EM, immunosuppressive therapy represents the first-line approach, including early administration of high-dose corticosteroids that provide a beneficial effect in preventing the progression of cardiac damage characterized by the development toward intermediate thrombotic necrosis and fibrosis stages with mural thrombosis [79]. Corticosteroids often relieve symptoms due to their potent anti-inflammatory effect. This strategy is also used in uncomplicated forms without a histological demonstration of myocarditis, involving systemic conditions such as EGPA, HES, and hypersensitivity EM [3,16]. However, despite the wide use of corticosteroids in non-EGPA-related EM, there is modest evidence in the medical literature based on small, non-randomized studies and case reports [80]. The initial dosage of corticosteroids and treatment duration vary among studies, making it difficult to provide clear, evidence-based recommendations. It is reasonable to adjust the dosage and treatment duration based on the severity of EM manifestations, including the degree of left ventricular dysfunction, myocardial necrosis marker levels, the primary underlying disorder, and the cardiac inflammation trends according to the results of the control biopsy and/or cardiac MRI during the follow-up. However, corticosteroids dramatically improved clinical symptoms, systolic dysfunction, and markers of inflammation and myocardial damage [81,82], suggesting that eosinophilic myocarditis could be an auto-immune disease.

In advanced stages of EM, other immunosuppressive drugs could be administered in addition to corticosteroids such as cyclophosphamide, methotrexate (above all EGPA-associated EM), azathioprine, hydroxyurea, or interferon-α (especially in steroid-refractory cases of HES) [3]. These strategies allow the reduction of steroid dosage and mitigate the iatrogenic effects of prolonged corticosteroid use, on top of their similar effectiveness compared to the single administration of high doses of corticosteroids. A study conducted by Miszalski-Jamka et al. demonstrated that patients with EGPA who early started non-corticosteroid immunosuppressive treatment had less new heart failure onset or progression compared to subjects in whom this therapy had not begun at diagnosis [83]. Albeit up to 35% of EM was idiopathic or undefined, it is possible to recognize a secondary cause in about 65% of the cases. The identification of EM-associated conditions is crucial for specific treatments, alone or in addition to corticosteroids: in hypersensitivity or allergic EM, potential causative factors must be identified and eliminated; targeted antimicrobial therapy is necessary for EM associated with helminthic infections (i.e., albendazole); tyrosine kinase inhibitors could be used in myeloproliferative disorders with eosinophilia characterized by *PDGFRA/B*-rearrangement; EGPA-related EM may benefit from cyclophosphamide therapy [16]. Furthermore, treatment could also involve monoclonal antibodies, including rituximab, which acts against CD20+ B cells, and mepolizumab, which inhibits the binding of interleukin-5 (IL-5) to its receptors expressed on eosinophils, improving cardiac function and reducing pericardial effusion [84]. Mepolizumab could be used as a supplement to steroid therapy in EM, including EGPA and HES, to reduce steroid doses and prevent their side effects [16]. In addition, new therapeutic strategies for eosinophilic-associated disorders have been suggested, including monoclonal antibodies directed against molecular targets involved in eosinophilic inflammatory pathways such as IL-5 receptor alpha, chemokine receptor CCR3, and sialic acid-binding immunoglobulin-like lectin 8 [78,85]. In recent studies, the humanized antibody to the IL-5 receptor benralizumab has been demonstrated to reduce blood and tissue eosinophilia [85,86]. However, further studies are needed to approve these new drugs targeting eosinophil receptors.

As mentioned above, EM could be associated with mural and intravascular thrombus, suggesting a hypercoagulable state. Specifically, endo-cavitary thrombi were reported in 12 of histologically proven EM and 28 of non-histologically proven EM, mainly in disorders with persistent eosinophilia (HES and EGPA) [3], as the damage induced by persistent eosinophilic could promote thrombotic and fibrotic transition from acute EM to Loeffler endo-myocarditis [87]. Indeed, these findings suggest the use of anticoagulation prophylaxis in the acute phase of EM for specific subgroups of patients to prevent the formation of thrombi. Regarding fulminant EM, the pharmacological therapy is based on drugs used for cardiogenic shock or acute heart failure, such as loop diuretics and inotropes, including dopamine, epinephrine, or noradrenaline, especially in forms with predominant systemic inflammation [3,4]. Moreover, the onset of ventricular arrhythmias requires the administration of intravenous amiodarone and the possible correction of electrolytes [4]. Non-pharmacological support may be necessary for EM with a fulminant presentation with impairment of left ventricle ejection fraction and malignant arrhythmias; these include respiratory support with non-invasive ventilation in the absence of marked hypotension or mechanical ventilation and mechanical adjuvant therapy such as ventricular assist devices, intra-aortic balloon counter-pulsation, and veno-arterial extracorporeal membrane oxygenation (ECMO) [76,88].

## 8. Prognosis

As mentioned above, myocarditis natural history could range from complete recovery to evolution in cardiomyopathy or death due to ventricular arrhythmias and/or severe systolic dysfunction [1]. Specifically, the prognosis in patients with eosinophilic myocarditis (EM) is hard to determine due to the scarcity of data and because it is closely related to the underlying etiology. One of the most extensive systematic reviews estimates an average in-hospital mortality of approximately 17.2%, reaching 31.4% in cases of eosinophilic myocarditis due to hypersensitivity [3]. However, an inevitable overestimation of mortality must be considered, as most pauci-symptomatic or asymptomatic EM cases go unrecorded [3]. Moreover, EM in fulminant presentation occurs with severe left ventricular dysfunction and high arrhythmic risk, in addition to increased mortality and the need for heart transplantation [77]. Indeed, circulatory support with inotropes or mechanical circulatory support (MCS) is necessary in many cases [77]. However, the use of inotropic therapy or MCS was not associated with worse outcomes in terms of recovery, as most patients showed a complete recovery of systemic function at the time of discharge [3]. After discharge, 30% of patients survive less than 3 years [16], but no specific registry defines the risks of recurrence, disease progression, or the likelihood of developing future ventricular arrhythmias.

## 9. Conclusions

Eosinophilic myocarditis (EM) is a rare inflammatory heart disease, often neglected, characterized by eosinophil-mediated damage and a high early mortality rate. Up to 25% of patients with eosinophilic myocarditis do not show peripheral eosinophilia, favoring the misdiagnosis. Moreover, EMB, the diagnostic gold standard, is widely underperformed and is not very sensitive, given the focal nature of cardiac inflammation. In addition, the tempestive identification of underlined etiologies is relevant for an early and specific treatment that could improve survival, as HES- and EGPA-related forms benefit from steroid therapy. Thus, the identification of “red flags” could lead to a prompt diagnosis of EM (Figure 5). Finally, clinical trials and international registries are needed to enhance knowledge about this disease, improving the acute and long-term prognosis.

## Figures and Tables

**Figure 1 biomedicines-12-00656-f001:**
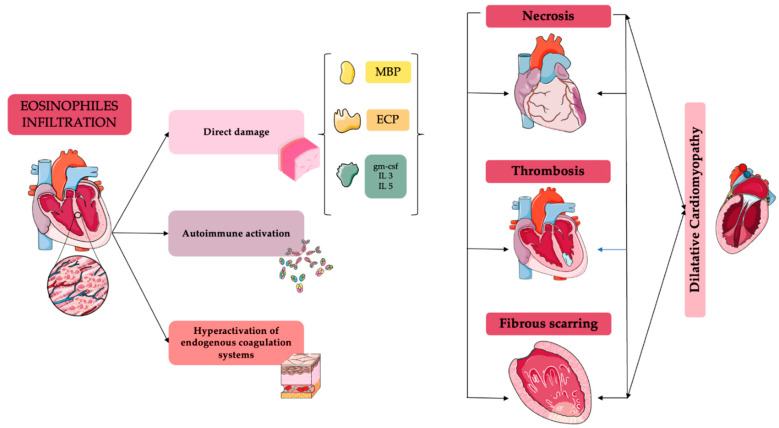
Pathophysiology and mechanism of damage in eosinophilic myocarditis. Abbreviations. ECP = eosinophilic cationic protein; gm-csf = granulocyte/macrophage colony-stimulating factor; IL-3 = interleukin 3; IL-5 = interleukin 5; MBP = major basic protein.

**Figure 2 biomedicines-12-00656-f002:**
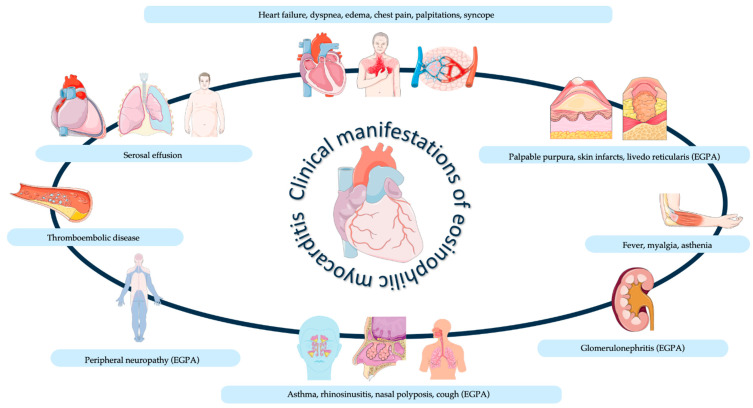
Clinical manifestations of eosinophilic myocarditis, related to different organs involved. Abbreviations. EGPA = eosinophilic granulomatosis with polyangiitis.

**Figure 3 biomedicines-12-00656-f003:**
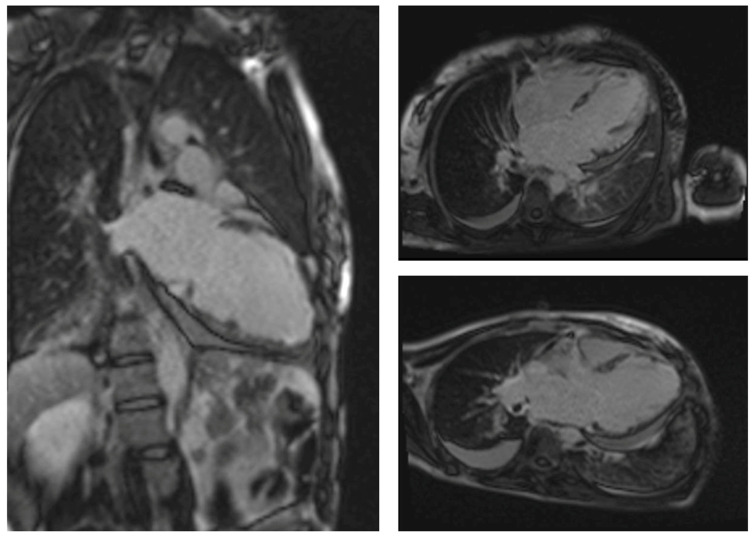
CMR findings in patients with EGPA-eosinophilic myocarditis. LGE images show sub-endocardial hyperintensity involving the mid-apical ventricular segments and focal areas of infra-myocardial hyperintensity in correspondence with the inferior septum in the basal and mid-apical regions.

**Figure 4 biomedicines-12-00656-f004:**
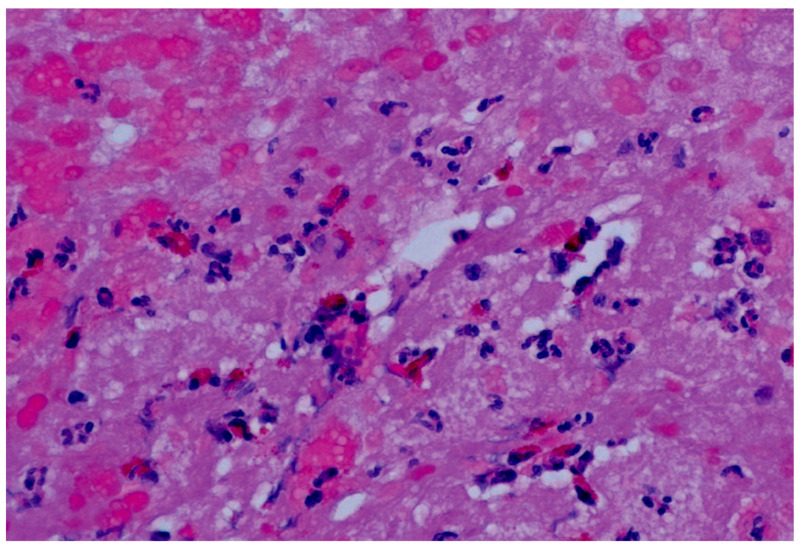
EMB in eosinophilic myocarditis. Hematoxylin and eosin staining shows eosinophilic infiltrate and diffuse myocardial damage with necrosis and subversion of myocardial tissue structure.

**Figure 5 biomedicines-12-00656-f005:**
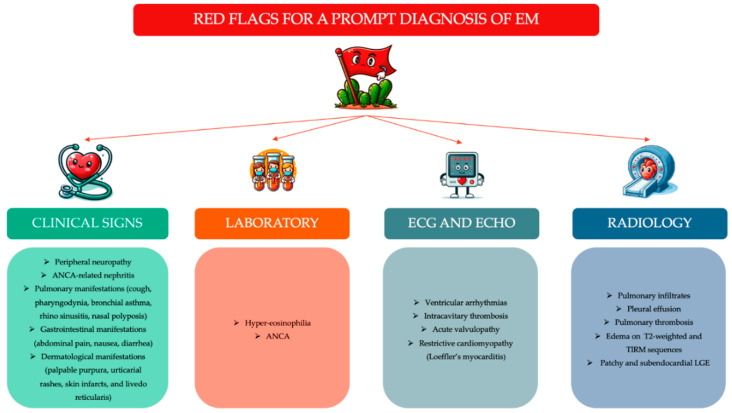
“Red flags” in eosinophilic myocarditis. Abbreviations. ANCA = anti-neutrophil cytoplasmatic antibodies; EM = eosinophilic myocarditis; LGE = late gadolinium enhancement.

**Table 1 biomedicines-12-00656-t001:** Etiologies of eosinophilic myocarditis.

Etiology	Associated Clinical Condition
Hypersensitivity [17,18,19,20,21,22,23,24,25,26]	○ Antibiotics (36.5%, mainly minocycline and β-lactam). ○ Central nervous system agents (21.1%, primarily clozapine and carbamazepine). ○ Anti-inflammatory (largely indomethacin). ○ Diuretics. ○ Vaccines (7.7%, mostly tetanus toxoid and smallpox). ○ Antitubercular agents (1.9%). ○ Prolonged exposure to high doses of dobutamine.
Vasculitis [27,28,29]	○ Eosinophilic granulomatosis with polyangiitis (EGPA).
Hyper-eosinophilic syndrome (HES) [5,18,31,32,33]	○ Davies’ endomyocardial fibrosis. ○ Loffler’s myocarditis. ○ Secondary to hematological disease.
Infections [3,5,34,35,36,37,38]	○ Parasites (*Toxocara canis*, *Trichinella spiralis*, *Entamoeba fragilis*, and *Isospora belli*). ○ Protozoa (*Trypanosoma cruzi* and *Toxoplasma gondii*). ○ Fungi (Aspergillus). ○ Viruses (HIV).
Cancer [39]	○ T-cell lymphomas. ○ Lung cancer. ○ Biliary cancer.
Other causes [40,41]	○ Pregnancy. ○ Peripartum.
Idiopathic [3]	○ The cause remains unidentified.

Abbreviations. EGPA = eosinophilic granulomatosis with polyangiitis; HIV = human immunodeficiency virus; HES = hyper-eosinophilic syndrome.

**Table 2 biomedicines-12-00656-t002:** Therapy in eosinophilic myocarditis.

**Context**	**Drugs**
Systolic dysfunction [4]	- Low-dose beta-blockers. - Angiotensin-converting enzyme inhibitors/angiotensin receptor blockers. - Aldosterone receptor antagonists. - Diuretics.
Pulse Therapy [3,16]	- Iv Methylprednisolone 7–14 mg/kg/day for three days.
Maintenance Therapy [3,16]	- Oral prednisone 1 mg/kg/day (tapered down progressively). - Immunosuppressive drugs (alone or in addition to steroid therapy): azathioprine, methotrexate, monoclonal antibodies (rituximab and mepolizumab), tyrosine kinase inhibitors (imatinib).
Fulminant EM [76,77]	- Loop diuretics. - Inotropes (dopamine, norepinephrine, and epinephrine). - Mechanical support of circulation (intra-aortic balloon counter-pulsation, ECMO, LVAD, etc.).
EGPA-related EM [16,78]	- i.v., cyclophosphamide (600 mg/m^2^ per day every two weeks for 30 days) - mepolizumab.
Infections [15,16]	- Albendazole (200–800 mg per day for 2–7 weeks) for helminthic infections (i.e., *Toxocara* spp.).
Hyper-eosinophilic Syndrome (HES) [3,16]	- *FIP1L1/PDGFRA*-associated HES: imatinib 100 mg daily. - *FIP1L1/PDGFRB*-associated HES: imatinib at starting dose of 400 mg daily lowered to 100 mg daily. - Mepolizumab. - Hydroxyurea or IFN-α (steroid-refractory cases of HES).
Hypersensitivity or Allergy [16]	- Potential causative factors must be identified and eliminated.

Abbreviations. ECMO = extra corporeal membrane oxygenation; EGPA = eosinophilic granulomatosis with polyangiitis; EM = eosinophilic myocarditis; HES = hyper-eosinophilic syndrome; IFN = interferon; IV = intravenous; LVAD = left ventricular assist device.

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
