# Peer review of "Eosinophilic Myocarditis: From Bench to Bedside"

_biomedicines, 2024, doi:10.3390/biomedicines12030656_

Round 1

Reviewer 1 Report

Comments and Suggestions for Authors

This nice well-written review provides an overview of eosinophilic myocarditis. The authors cover all the aspects of the disease in a comprehensive way. There are a few comments which may improve reading and the value of the review.

In the diagnostic section the various modalities namely ECG, echo, blood tests, CMR are presented by the authors as equivalent for the diagnosis of EM. However, CMR has revolutionized the diagnostic workout of myocarditis and provides a quick and validated diagnosis is patients with variety of symptoms ranging from asymptomatic to life threatening conditions. For example patients with vasculitis may be oligo and/or asymptomatic, have normal EGC, no echo abnormalities and significant myocardial involvement (J Cardiovasc Magn Reson. 2017 Jan 6;19(1):6. Autoimmun Rev. 2012 Dec;12(2):305-12). In my opinion this should be stressed and discussed in the relevant section.

Similarly, the subtle clinical signs of EM in the setting of EGPA may be overlooked due to coexistence of the systemic features of the disease and this is another important point highlighting the role of CMR not only in the diagnosis but also in the pathophysiology of cardiac involvement in EGPA. (Mediterr J Rheumatol. 2021 Mar 31;32(1):15-20,  Int J Cardiovasc Imaging. 2021 Apr;37(4):1371-1381).

It also appears that CMR may be an important player in risk stratification and treatment decisions in EM associated wit EGPA (Rheumatology Oxford. 2024 Feb 9:keae085). In that respect the monitoring of treatment response including blood tests, echo, CMR could be discussed (Vasc Diffuse Lung Dis 2016; 33:51 – 58, Curr Opin Rheumatol. 2019 Jan;31(1):16-24,).)

There is a significant overlap between etiology and epidemiology sections particularly in view of the cause of EM. The authors could revise it accordingly by shortening the extend of etiology part.

A couple of nice echo, CMR pictures may enhance the paper.

Author Response

This nice well-written review provides an overview of eosinophilic myocarditis. The authors cover all the aspects of the disease in a comprehensive way. There are a few comments which may improve reading and the value of the review.

We appreciate the positive feedback from the reviewer, and we thank him/her for all the many helpful comments that we believe to have fully addressed in this revised version of the manuscript.

In the diagnostic section the various modalities namely ECG, echo, blood tests, CMR are presented by the authors as equivalent for the diagnosis of EM. However, CMR has revolutionized the diagnostic workout of myocarditis and provides a quick and validated diagnosis is patients with variety of symptoms ranging from asymptomatic to life threatening conditions. For example patients with vasculitis may be oligo and/or asymptomatic, have normal EGC, no echo abnormalities and significant myocardial involvement (J Cardiovasc Magn Reson. 2017 Jan 6;19(1):6. Autoimmun Rev. 2012 Dec;12(2):305-12). In my opinion this should be stressed and discussed in the relevant section.

Similarly, the subtle clinical signs of EM in the setting of EGPA may be overlooked due to coexistence of the systemic features of the disease and this is another important point highlighting the role of CMR not only in the diagnosis but also in the pathophysiology of cardiac involvement in EGPA. (Mediterr J Rheumatol. 2021 Mar 31;32(1):15-20,  Int J Cardiovasc Imaging. 2021 Apr;37(4):1371-1381).

We thank the reviewer for his/her comment. We added more information regarding the utility and the crucial role of CMR in pathophysiology and diagnostic workout of EM, especially in patients with EGPA (Page 3, Lines 120-127; Page 8, Lines 319 – 328; Page 9, Lines 329-344).

It also appears that CMR may be an important player in risk stratification and treatment decisions in EM associated wit EGPA (Rheumatology Oxford. 2024 Feb 9:keae085). In that respect the monitoring of treatment response including blood tests, echo, CMR could be discussed (Vasc Diffuse Lung Dis 2016; 33:51 – 58, Curr Opin Rheumatol. 2019 Jan;31(1):16-24,).)

We thank the reviewer for the comment. We think that it’s a very intriguing point, as CMR could be useful in monitoring disease and response to therapy, beyond its well-known importance in diagnosis. In this regard, we added information in the “diagnosis” section (Page 9, Lines 345-351).

There is a significant overlap between etiology and epidemiology sections particularly in view of the cause of EM. The authors could revise it accordingly by shortening the extend of etiology part.

We thank the reviewer for the comment. We revised the etiology paragraph, removing overlaps with epidemiology paragraph.

A couple of nice echo, CMR pictures may enhance the paper.

We thank the reviewer for his/her suggestion. We added a new figure (Figure 3) which showed LGE images of a CMR performed in a patient with EGPA-related EM.

Reviewer 2 Report

Comments and Suggestions for Authors

Piccirillo and colleagues conducted a review of literature on eosinophilic myocarditis, a rare subtype of myocarditis with a high reported early mortality rate that is often neglected by physicians and cardiologists. This review comprehensively described the epidemiology, etiology, pathophysiology, clinical manifestations, diagnosis, therapy, and prognosis of EM. In general, this is a well-written review with updated information. Below are some minor concerns:

1. Since the title of the review is “Eosinophilic myocarditis: from bench to bedside”, could the authors add more basic or translational studies on the etiology or pathophysiology or treatment of EM?

2. In Section 2. Epidemiology, several studies estimated the presence of EM in 20% of hearts explanted for transplantation, which is quite impressive. Could the authors cite original studies supporting such data?

3. Some abbreviations are not properly used. For example, EM and eosinophilic myocarditis were mixed used throughout the manuscript.

Author Response

Piccirillo and colleagues conducted a review of literature on eosinophilic myocarditis, a rare subtype of myocarditis with a high reported early mortality rate that is often neglected by physicians and cardiologists. This review comprehensively described the epidemiology, etiology, pathophysiology, clinical manifestations, diagnosis, therapy, and prognosis of EM. In general, this is a well-written review with updated information. Below are some minor concerns:

We appreciate the positive feedback from the reviewer, and we thank him/her for all the many helpful comments that we believe to have fully addressed in this revised version of the manuscript.

Since the title of the review is “Eosinophilic myocarditis: from bench to bedside”, could the authors add more basic or translational studies on the etiology or pathophysiology or treatment of EM?

We thank the reviewer for the suggestion. We added more information regarding pathophysiology (Page 5, Lines 189-190 and Lines 199-203; Page 6, Lines 204-205)

In Section 2. Epidemiology, “several studies estimated the presence of EM in 20% of hearts explanted for transplantation”, which is quite impressive. Could the authors cite original studies supporting such data?

We thank the reviewer for the comment. We agree on the impressive rate of EM in transplanted hearts: beyond direct damage, we could hypothesize a correlation with long-term use of dobutamine, which is recognized as a cause of EM. We added references as suggested (Page 2, Lines 58).

Some abbreviations are not properly used. For example, ‘EM’ and ‘eosinophilic myocarditis’ were mixed used throughout the manuscript.

We thank the reviewer for the suggestion, as it could be confusing. Thus, we used “eosinophilic myocarditis” as first appeared in each paragraph and the abbreviation “EM” in the subsequent text.

Round 2

Reviewer 1 Report

Comments and Suggestions for Authors

The authors have adequately addressed the comments.

However a couple of suggested references (Mediterr J Rheumatol. 2021 Mar 31;32(1):15-20, Curr Opin Rheumatol. 2019 Jan;31(1):16-24,) )should be included to enrich the reference list

Author Response

The authors have adequately addressed the comments.

However a couple of suggested references (Mediterr J Rheumatol. 2021 Mar 31;32(1):15-20, Curr Opin Rheumatol. 2019 Jan;31(1):16-24,) )should be included to enrich the reference list

We appreciate the positive feedback from the reviewer. We added the suggested references (n. 69 and 74) to improve the quality of manuscript.